# Investigation on *Phoenix dactylifera/Calotropis procera* Fibre-Reinforced Epoxy Hybrid Composites

Mohammad Hassan Mazaherifar [1], Hamid Zarea Hosseinabadi [1], Camelia Coșereanu [2,*], Camelia Cerbu [3,*], Maria Cristina Timar [2] and Sergiu Valeriu Georgescu [2]

1  Department of Wood and Paper Science and Technology, Faculty of Natural Resources, University of Tehran, Karaj 31585-4314, Iran
2  Faculty of Furniture Design and Wood Engineering, Transilvania University of Brasov, B-dul Eroilor nr. 29, 500036 Brasov, Romania
3  Department of Mechanical Engineering, Faculty of Mechanical Engineering, Transilvania University of Brasov, B-dul Eroilor nr. 29, 500036 Brasov, Romania
*  Correspondence: cboieriu@unitbv.ro (C.C.); cerbu@unitbv.ro (C.C.)

**Abstract:** This paper presents the investigations conducted on three types of fibre-reinforced epoxy-resin hybrid composites with different structures, manufactured using midrib long fibres of date palm (*Phoenix dactylifera* L.) and *Calotropis procera* fibres. The two types of fibres were formed into flat sheets, without adding other chemicals or resins, and employed as reinforcing layers in the structure of the multi-layered laminate composites. Three-layer and five-layer epoxy-reinforced laminates were manufactured from the sheets of date-palm fibres and *Calotropis* sheets bonded with laminar epoxy resin. Water resistance investigation and mechanical testing under tensile, bending and impact loads were conducted in the research in order to evaluate and compare the performance of the resulting composites. Emphasis was put on the effect of various factors, such as the type of reinforcement material and the number of plies in the laminate on the mechanical behavior of the composites. The interpretation of those results was supported by the stereo-microscopic investigation of the adhesion between the layers of the composites, and the vertical density profile (VDP), which showed the repartition of the density on the composite thickness depending on the layer material. The results of the mechanical performance of the composites showed lower values of tensile strength, tensile modulus of elasticity and impact resistance and an increase of water absorption (WA) and thickness swelling (TS) for the five-layer composites compared to the three-layer composites. Contrarily, the addition of *Calotropis* fibres improved the flexural strength and the flexural modulus of elasticity. The alkali treatment of the *Calotropis* fibres improved the mechanical performance of the composites compared to the ones made with untreated fibres, because of an apparent increase in cellulose content and free hydroxyl groups revealed by FTIR spectra.

**Keywords:** date palm; long fibres; *Calotropis* fibres; epoxy resin; laminate composites; mechanical testing

## 1. Introduction

Composite materials are used as alternatives to conventional materials in industries such as automotive, aerospace and buildings because of their improved characteristics including higher mechanical strengths and reduced specific weight. The increasing demand for sustainable and renewable materials brought to the attention of specialists the possibility of using natural lingo-cellulosic fibres in several applications, correlated to their light weight and high strength. On the other hand, the employment of various natural lingo-cellulosic fibres instead of wood fibres reduces the forest trees' exploitation rate. One of the renewable resources with high potential for use is date palm (*Phoenix dactylifera* L.), which is widely cultivated in the Middle East and North Africa for its fruit crops, covering important areas in these regions [1,2]. Saudi Arabia, Algeria, Iran, Iraq, and Egypt have the highest palm ranks

in the world [3] and the area under cultivation indicates a percentage of 21% of the world's date-palm groves belonging to Iran. The world's total number of date palms is more than 120 million [4]. Approximately 11.8 million hectares are under cultivation for date-palm trees, distributed in 94 countries. Overall, the date-palm waste, generated from seasonal pruning and trimming, can be estimated at an average of 35 kg per tree [5]. Considering the 120 million date-palm trees, and 35 kg of waste per tree, 4,200,000 tons of natural fibres can be consumed in various fields, a fact which helps to prevent cutting forest trees around the world. Consequently, these residues, which are renewable, have good potential to be used in the composite production industry [6], such as for the reinforcement of polymer composites for automotive or maritime industries, for construction as geotextiles, and the reinforcement of asphalt concrete or gypsum plaster [2,7].

Long fibres extracted from the midrib of the date palm and spanned into yarns and further alkaline treated have physical, chemical, morphological and mechanical properties comparable to those of other natural fibres, such as sisal, hemp, and flax [7]. A tensile strength of date-palm midribs of 11.4 Mpa, higher than for bamboo or sisal fibres, was reported in the literature [8]. Date-palm fibres can be obtained from annual pruning by-products such as spadix stems, midribs, and leaflets [9], but the longest ones can be obtained only from midribs, which contain them in their natural matrix [7].

*Calotropis procera* is a small perennial tree of the *Apocynaceae* family and it is native to Africa, the Arabian Peninsula, Western Asia, the Indian subcontinent and India, and was also introduced in South Africa, Australia, Latin America and the United States because of its economic benefits [10–14]. Commonly known as the giant milkweed, apple of sodom, or calotrope [13], this plant grows slowly, and it is drought-resistant and evergreen with softwood, having thick branches that may grow up to 6 m. The fruits split at maturity to release numerous seeds, around 350–500 seeds per fruit [15], with bundles of white silk or Papus fibres having several applications in the chemical industry, building industry and medicine [16–18]. Apart from these applications, Papus fibre is a natural and renewable material, composed of 64.0 wt% of cellulose, 19.5 wt% hemicellulose, and 9.7 wt% lignin, being a source of ligno-cellulosic fibres with low density and high strength [13,19]. These fibres are lightweight hollow tubes [20–22] with thin walls and low density, but high strength and hydrophobic properties, being very suitable not only as insulation materials but also as natural fibres for reinforced composites [20]. An alkaline treatment applied to the *Calotropis* fibres extracted from the stem can improve the mechanical properties when they are used in reinforced epoxy polymer composites. This was explained as a result of the increase of the cellulose content and decrease of the fibre density following alkaline treatment, resulting in effective bonding at the fibre–matrix interfaces [23].

As reinforcements in composites, fibres have to have high strength, high stiffness, and low density, whilst the matrix requires good shear properties. Carbon fibres, glass fibres, and aramid fulfill these requirements and are preferred as reinforcement materials of the advanced composites [24–26], and they are being used in the automotive industry for automobile bodies, civil engineering applications for strengthening walls, or in maritime applications for ship hulls [27]. Natural fibres are studied by many researchers as alternative materials for synthetic fibres in the industry of composite materials, due to their low cost and lightweight properties, bringing important advantages when used in polymer matrix composites. A study on 2D woven kenaf fibre-reinforced acrylonitrile-butadiene-styrene (ABS) has shown that the alkali treatment of the kenaf woven fabric led to the increase of the adhesion between fibres and ABS, with a beneficial effect on the tensile strength of the composite [28]. Generally, the drawback of the natural fibres is their low resistance to water, which is why it is preferred to use them for applications in the indoor environment. In this context, date-palm fibres represent an agricultural waste suitable as a reinforcement for polymeric composites used in the automotive industry as an interior component [29]. Alkali treatment applied to natural fibres increased the compatibility between the fibres, and polymers in fibre-reinforced polymer composites, improving the mechanical performance

of the resultant structures [30–32]. This effect was explained by wax removing and the fact that the fibres tend to be densely packed due to the removal of hemicellulose [32].

In the present study, three types of multi-layered laminate composites with different structures were developed and manufactured, employing long date-palm (*Phoenix dactylifera* L.) fibres extracted from the midribs and *Calotropis* white silk fibres as reinforcing materials, and epoxy resin as the matrix. The envisaged applications of these composites are in the automotive, aerospace and construction industries, as potential alternatives to the synthetic fibre-reinforced composites, which are more expensive and require complex manufacturing technologies. The advantage of using these natural fibres is the availability of a large amount of waste, their low cost and low $CO_2$ footprint. The research conducted on the developed composites focused on the effect of some factors, such as the type of reinforcing material, the alkali treatment of *Calotropis* silk, and the number of plies in the laminate on the mechanical properties of the composites. For a more comprehensive study, a stereo-microscopic analysis of the fibres and of the adhesion between layers, an investigation of the vertical density profile (VDP), and an FTIR analysis of *Calotropis* fibres before and after alkaline treatment and resistance to water were conducted in order to assist in the interpretation of the mechanical tests results.

## 2. Materials and Methods

### 2.1. Materials

2.1.1. Date Palm Midrib Long Fibre Extraction

Based on the anatomic features of date-palm parts, just the midrib consisted of long fibres positioned in the natural matrix [7]. The entire process of the fibre extraction following the procedure used by other researchers [7] is presented in Figure 1.

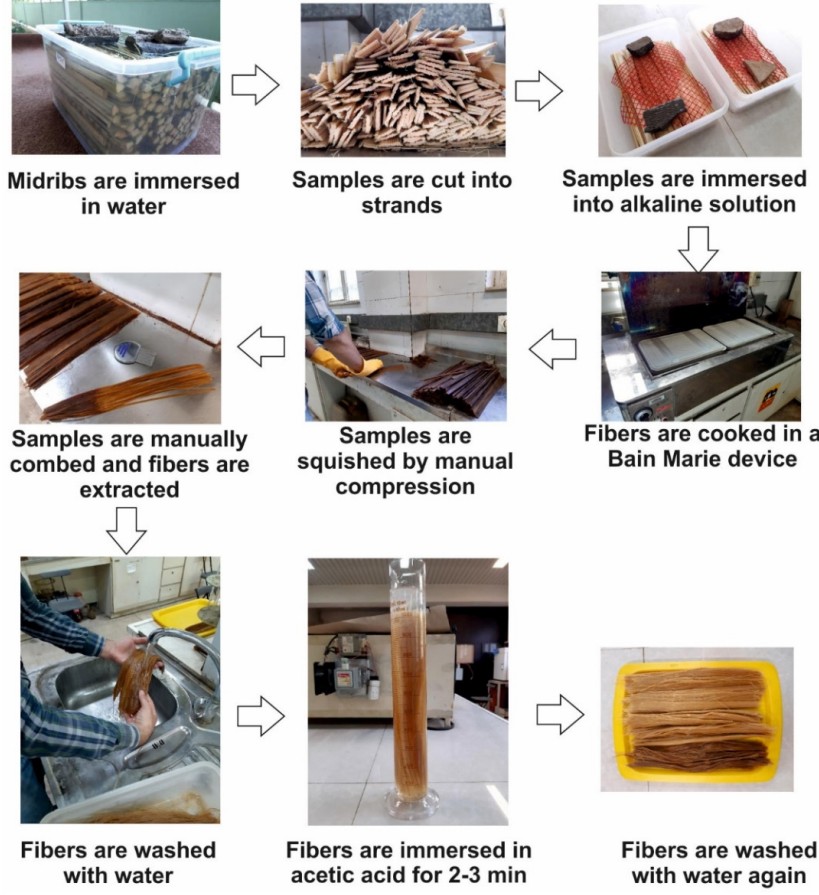

**Figure 1.** An illustration of the process of date-palm midrib long fibre extraction.

To provide the raw materials needed for extraction of the date-palm fibres, date-palm midribs were obtained from the date-palm groves of Hormozgan province (Rudan city), Iran. The middle parts of the midribs were used to extract the fibres. After cutting the midribs at the length of 400 mm, their moisture content was increased by immersing the samples in water for seven days and changing the water at least once every 24 h.

After seven days of immersion in water, the samples were cut manually into thinner strands with a knife or blade, and the outer skin of the midrib was removed from each strand, so to allow a better penetration of sodium hydroxide into the core. The fibres close to the outer skin of the midribs are stiffer fibres than the fibres of the middle parts, so when the samples were converted to strands, these fibres were removed. Then, the strands were immersed in an alkaline solution of sodium hydroxide 1.5% (ratio of 1:20 as strand: NaOH solution) and cooked in a Bain Marie device at a temperature of 95 °C for three hours. The NaOH solution was used for delignification. Then, the samples were squished by manual compression with rollers. During this process, the lignin and extracts that were softened during the cooking process were squished, and the fibres were separated. The fibres were afterwards separated with a metal comb with a very small distance between the teeth, and we performed this manual combing operation until all the waste was removed from the fibres. After extracting the fibres, in the last step, a neurulation of alkaline fibres with acetic acid (concentration of 5%) by immersion for 2–3 min was performed. First, the fibres were washed with water so that they were completely clean, and this operation was repeated after the immersion into the acid, so finally, the color of the fibres was brighter. The washing steps with water were conducted to produce cleanliness and further delignification.

The extracted date-palm fibres were longitudinally oriented and arranged to form flat sheets for stratified composite layers. The date-palm midrib long fibres were put in water until they converted to flat sheet form, then they were longitudinally oriented and pressed by hand and left for 24 h to dry under normal environmental conditions (temperature of 20 °C and 55% relative humidity of the air). The date-palm sheets formed this way had sizes (length × width) of about 350 mm × 350 mm.

The midrib long fibres of date palm were used in the composition of three-layer laminate (D), with adjacent layers having perpendicularly oriented fibres, in the basic structure of the five-layer composites DC and DTC (Table 1).

**Table 1.** The structure of experimental composites.

| Code | Date Palm Fibres | *Calotropis* Fibres |
|---|---|---|
| D | 3 layers | - |
| DC | 3 layers | 2 layers (untreated) |
| DTC | 3 layers | 2 layers (treated) |

2.1.2. Calotropis Fibres Extraction and Treatment

To provide the fibres needed for *Calotropis* sheets, the mature fruits (Figure 2a) were collected from Sistan and Baluchestan province (Iran), and the white silk (Papus fibres) were separated from the seeds (Figure 2b).

The *Calotropis* sheets with sizes (length × width) of 350 mm × 350 mm were formed by arranging the fibres as seen in Figure 2c. The sheet that formed was sprinkled with water and left for drying for 24 h in an indoor environment, at a temperature of 20 °C and 55% air relative humidity. Without being pressed, the fibres connected to one another and formed a continuous sheet weighing approximately 3 g. These sheets were used to manufacture the five-layer composite (DC), as shown in Table 1.

The alkali treatment of *Calotropis* sheets has the role of removing unwanted substances such as oil or wax from the fibres' surfaces, providing improved mechanical properties due to the increased amount of cellulose [23]. First, the fibres were immersed for 30 min in a 5% solution of sodium hydroxide (NaOH) prepared with distilled water, in order to remove non-cellulosic impurities from the fibres.

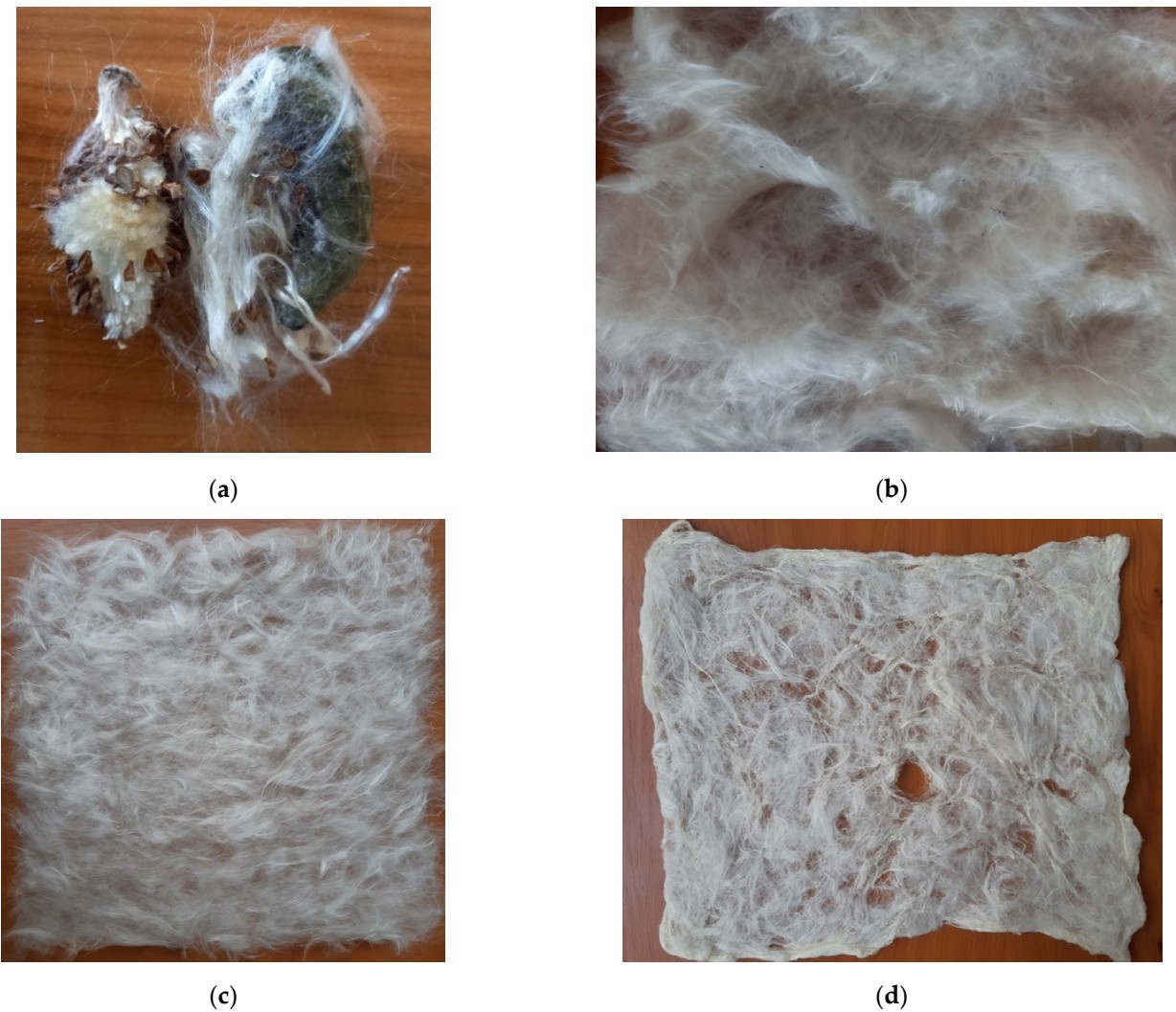

(a)

(b)

(c)

(d)

**Figure 2.** *Calotropis procera* fibres: (**a**) mature fruits; (**b**) extracted fibres arranged to form a thin sheet for composite layer; (**c**) untreated *Calotropis* sheet; (**b**) treated *Calotropis* sheet.

In the next step, the fibres were washed with distilled water to remove the sodium hydroxide solution, and after that, the fibres were immersed in dilute hydrochloric acid (HCl) solution for 1 min. The fibres were washed again with distilled water, and after drying, they were arranged to form a uniform rectangular layer of approximately 350 mm × 350 mm for composite manufacturing (Figure 2d). These treated *Calotropis* sheets were used in the manufacturing of five-layer composites (DTC) (Table 1).

*2.2. Composites Manufacturing*

Multi-layered fibre-reinforced epoxy composites were manufactured. The date-palm fibres and *Calotropis* fibres were formed into entangled sheets as previously presented. The Laminar BK epoxy resin, manufactured by Vosschemie GmbH company located in Uetersen, Germany, was used as a binder between the layers of fibres.

Three types of composites (Table 1) were manufactured under laboratory conditions: a three-layer composite made only from date-palm fibre layers (two longitudinally oriented layers for the faces and one transversally oriented for the core) (composite D) and five-layer composites made from three layers of date-palm fibre sheets and two layers of *Calotropis* sheets (one variant with untreated *Calotropis* fibres coded as DC and one variant with treated ones, coded as DTC). The moisture content of date-palm fibres was 3% before manufacturing the composites.

Epoxy BK is a two-component epoxy resin, is solvent-free, clear and transparent, and can be used as a laminating resin. The mixing ratio of the two components (A/B parts weight) is 100/60 and the hardness index of the resin is Shore D-80. The densities of the two components are 1.15 g/cm$^3$ for component A and 1 g/cm$^3$ for component B. The working temperature of the surface on which BK epoxy resin was applied was about 20 °C, and the working time for applying it in the structure of one composite panel was up to 30 min. According to the technical sheet, the BK epoxy resin reaches its final strength after 3–5 days.

The resin was applied by brush on each individual layer for fibres impregnation (Figure 3a). Each layer of the composite was firstly weighed with an accuracy of 0.01 g, and the epoxy resin amount was calculated by multiplying the weight of each layer as follows: five times for date-palm fibres and seven times for *Calotropis* fibres. The core layer of the date-palm composite was placed to ensure that the direction of fibres was perpendicular to those of the face layers (Figure 3b). White silicone baking paper was used at the top and bottom of the sandwich composite in order to avoid adhesion to the top and bottom supports (Figure 3c). The formed sandwich composite covered on both faces by silicone baking paper was placed between two blockboard panels, and a weight of 20 kg was placed on the top panel for cold pressing the composite layers at a pressure of 0.019 bar for three days. The composites were then conditioned for six days at a temperature of 20 °C and 55% air relative humidity before testing. Three replicates of each composite type (D, DC, DTC) were manufactured. After six days, the panels were sized at 300 mm × 300 mm. The final composite panels after sizing presented a compact appearance and a rigid structure (Figure 3e). Testing samples were then cut according to the specific standards requirements for measuring vertical density profile, water absorption, flexural strength and flexural modulus of elasticity, tensile strength and tensile modulus of elasticity, and impact resistance. The number of samples and the methods of testing were in accordance with the corresponding technical specifications, as further detailed in the respective sub-sections.

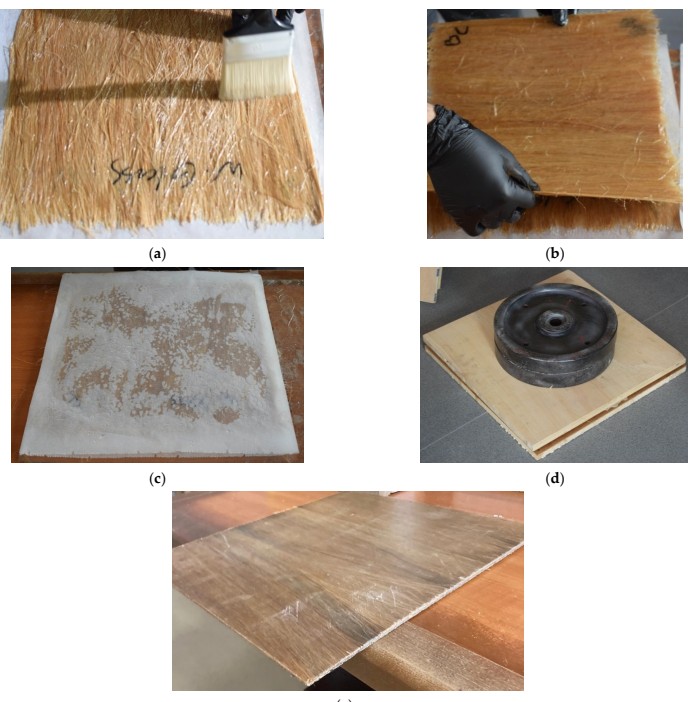

**Figure 3.** Composite manufacturing: (**a**) impregnation of the first layer with BK epoxy resin; (**b**) arrangement of middle date-palm layer; (**c**) composite covered by white silicone baking paper; (**d**) composite placed between two block board panels and pressed by two calibrated weights of 10 kg; (**e**) final composite panel.

### 2.3. Vertical Density Profile (VDP)

The vertical density profile (VDP) was investigated using the X-ray density profile analyzer DPX300 (IMAL, San Damaso, Italy). Five square-shaped specimens cut from each composite panel type were tested. The specimens had dimensions (length × width) of 50 mm × 50 mm, and the density profile was measured along the entire thickness of the sample. The specimens were first weighed with an EU-C-LCD precision scale (Gibertini Elettronica, Novate Milanese, Italy) and their dimensions were measured by the density profile analyzer.

### 2.4. Microscopic Investigation

The stereo-microscope NIKON SMZ 18-LOT2 (Nikon Corporation, Tokyo, Japan), was employed in the microscopic investigation and measurement of the fibres used for the composites, and also for better visualizing the structures of the manufactured composites and the adhesion between the component layers. In order to perform the measurements of the fibres' diameters, images with 240× magnification were taken. For the composites, the microscopy was performed on the edge of the structures with 22.5× magnification, and the constituted layers and the adhesion area between them were highlighted.

### 2.5. FTIR Analysis

Fourier transform infrared spectroscopy (FTIR) was employed to investigate the chemical features of the untreated *Calotropis* fibres, considered as control, as well as the chemical changes brought about by the alkaline treatment applied to three samples of *Calotropis* fibres. An ALPHA Bruker spectrometer (produced by Bruker Optik GmbH, Ettlingen, Germany) equipped with an ATR (attenuated total reflection) module was used to record the FTIR spectra in the range 4000–400 $cm^{-1}$ at a resolution of 4 $cm^{-1}$ and 24 scans/spectrum. Three individual spectra were recorded for each sample. The recorded spectra were further processed employing OPUS software for baseline correction and smoothing, and an average spectrum was calculated for each type of sample. After the average spectra were normalized (Max-Min normalization), they were compared to reveal chemical changes occurring as a result of the treatment applied. The characteristic absorption bands were assigned according to the references in the literature.

### 2.6. Water Immersion

Samples with dimensions (length × width) of 50 mm × 50 mm were cut from the composite panels in order to determine the thickness swelling and water absorption by immersion in water according to the SR EN 317: 1996 standard [33]. The test samples (five replicates for each type of composite) were immersed in a water bath at a temperature of 20 °C for 24 h. The sizes of the samples were measured by an electronic caliper with an accuracy of 0.01 mm. The samples were weighed before starting the test and after 24 h of immersion into the water using an electronic scale with the accuracy of 0.01 g. The thickness of the samples was measured every time at the diagonal cross point. The values were recorded for structures D and DC, investigating whether the layer of *Calotropis* fibres has an influence or not on the water absorption (WA) and on the thickness swelling (TS).

### 2.7. Mechanical Performance

The testing method and the number, shape and dimensions of the specimens used for each mechanical test performed were according to the corresponding European standards. Flexural strength and flexural modulus of elasticity values were determined according to the EN 310:1993 standard [34] and the equipment used for the test was the Zwick/Roell Z010 universal testing machine (Ulm, Germany).

Tensile tests were carried out on the universal testing machine LFV50-HM, 980 (Walter and Bai, Switzerland), which is digitally controlled. The maximum force provided by the machine is 200 kN. The tensile specimens were cut from the composite panels and their dimensions were established according to the European standard EN ISO 527-4 [35]. The

loading speed was 1.5 mm/min according to the same standard. The data recorded every 0.1 s were: tensile force *F*, elongation $\Delta l$ of the tensile specimen, and time *t*. The tensile modulus of elasticity and tensile strength were determined according to the standard EN ISO 527-4 [35]. The tensile modulus of elasticity was determined on the linear portion of the stress–strain curve recorded for each specimen.

The impact strength was investigated by the Charpy test, carried out on the pendulum impact tester HIT50P manufactured by Zwick/Roell (Ulm, Germany). The maximum capacity of the impact pendulum HIT50P with digital controlling is 50 J for the impact energy. The impact test set-up and the dimensions of 100 mm × 10 mm for the rectangular specimens were in accordance with the European standard ISO 179-1 [36]. The thickness of the specimens is provided by the thickness of the panel from which the specimens were cut. The dimensions of the cross-sections were accurately measured before testing for each specimen by the electronic caliper with the accuracy of 0.01 mm. In the Charpy impact test, the impact hammer hits the middle of the specimen, which is simply supported at both ends, and the elastic failure energy *W* is displayed digitally by the pendulum impact tester HIT50P, as in other experimental research works [37,38]. The impact strength or resilience denoted with *K* is computed as the ratio between the elastic failure energy *W* and the area *A* of the cross-section for each specimen by using Equation (1):

$$K = W/A \tag{1}$$

*2.8. Statistical Analysis*

The statistical analysis employed the determination of standard deviation in Microsoft Excel for a confidence interval of 95% and a significance level of 0.05 ($p < 0.05$). Single-factor Analysis of Variance (ANOVA) was performed with the Microsoft® Excel package for analyzing the way in which the average values of tensile and flexural strengths, modulus of elasticity (for bending and tensile loads), impact resistance, water absorption (WA), and thickness swelling (TS) were significantly affected by the participation of both treated and untreated *Calotropis* fibres in the structure of the composites and also by the direction of the fibres during the test (longitudinal and transversal). The direction of the fibres was considered to be the direction that the fibres in the composite face. Furthermore, Tukey's test was utilized to specify the significant differences created by the treatments.

## 3. Results and Discussion

*3.1. Vertical Density Profile (VDP)*

In Figure 4a–c, vertical density profile examples for each type of manufactured composites are presented.

The measured thicknesses for the three types of composite laminates varied as follows: between 6 mm and 6.77 for the D structure, between 6.43 mm and 7.4 mm for the DC structure, and between 6.82 mm and 7.42 mm for the DTC structure.

As seen in Figure 4a, minimum densities were recorded for positions of approximately 2.2 mm and 4.3 mm on the thickness of the composite, measured from the face. These minimum points on the graph correspond to the epoxy resin layers on the thickness, for which the calculated density of the mixture of the two components was 1094 kg/m$^3$. The maximum recorded densities were 1174.1 kg/m$^3$ for the left peak and 1208.7 kg/m$^3$ for the right peak, positioned at the thickness of 2.35 mm and 4.55 mm from the face of the composite and corresponding to the date-palm fibres layers (core and face layer). The vertical density profile clearly shows the structure of the composite, as follows: the three zones with low density variations and containing the maximum peaks belong to the date-palm layers (with thicknesses of approximately 2 mm), whilst the two minimum density points indicate the position of the epoxy resin layers. The recorded ratio between the minimum density and the average density was 0.99.

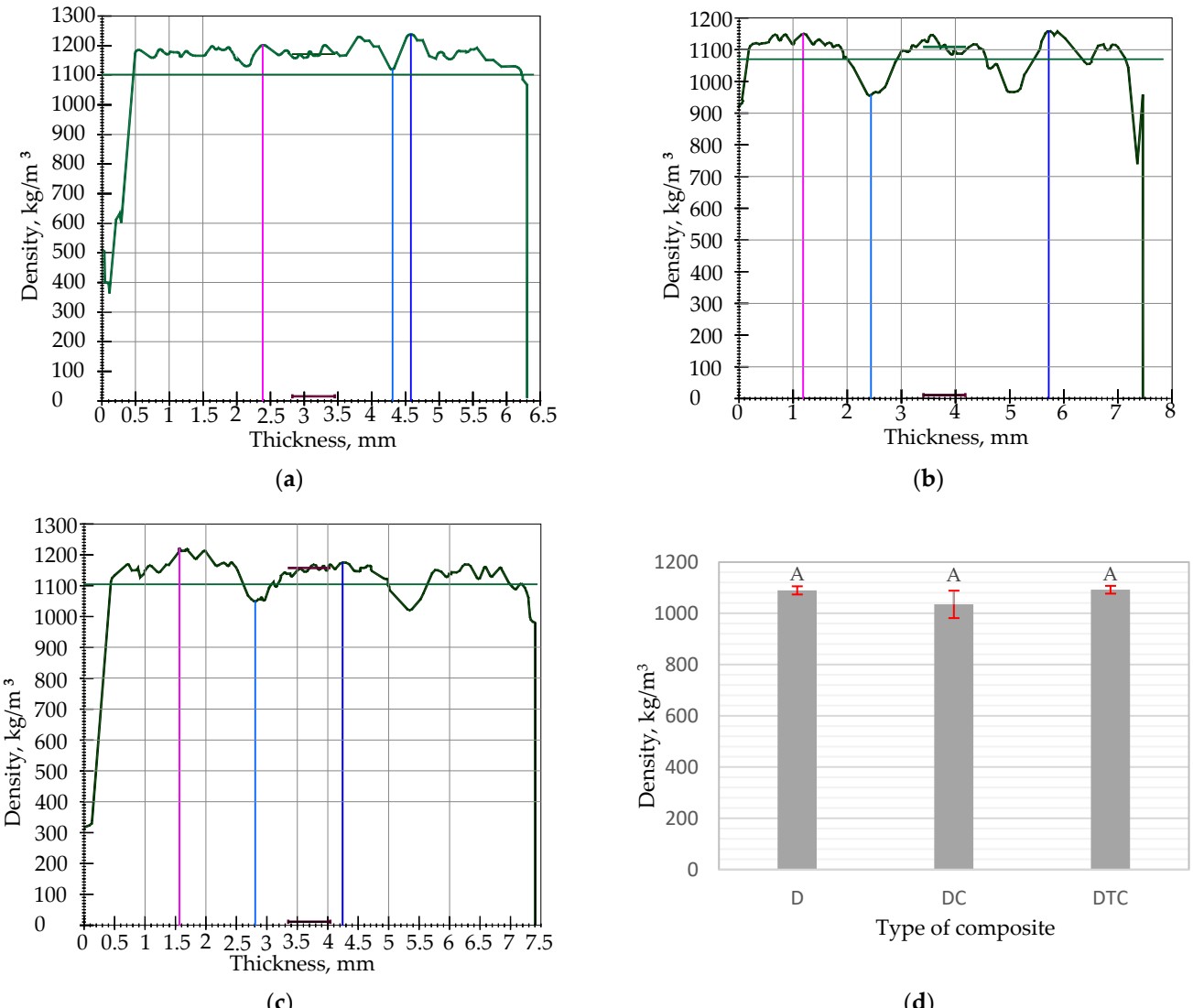

**Figure 4.** Vertical density profiles of the tested composites: (**a**) D; (**b**) DC; (**c**) DTC; (**d**) average values of measured densities.

The VDP of the DC structure shown in Figure 4b indicates higher differences between minimum points and maximum peaks of the graph. The minimum density recorded for this structure was 956.3 kg/m$^3$, located at the thickness of 2.45 mm from the outer face, indicating the presence of the *Calotropis* fibres layer, which had a density of approximately 0.025 kg/m$^3$ before pressing into the composite structure. The recorded ratio between the minimum density and the average density was 0.89, which was lower for the D structure. The VDP for the DC structure shows the layer's position along the thickness of the composite, with small variations of the densities corresponding to the date-palm layers, similar to the findings of VDP for the D structure.

In the case of the DTC composite, when treated *Calotropis* fibres were used, the VDP graph (Figure 4c) shows smaller differences between maximum density peaks and minimum points than in the case of using untreated *Calotropis* fibres (structure DC). However, the presence of the treated *Calotropis* layers is also noticed through the visible difference of densities between the maximum and minimum values. Instead, the maximum density peak is higher than for structure D, recording a value of 1220 kg/m$^3$. The ratio between the minimum density and the average density was 0.96 in this case, closer to the value calculated for the D structure.

The results show that the treatment applied to the *Calotropis* fibres affects the density of the composite, lowering the differences in densities between the *Calotropis* fibre layers and date-palm layers. This observation is also proved by the graph from Figure 4d, which shows a lower average density value for structure DC compared to the other two. The findings of the present investigation are in line with the results obtained by other researchers [2], who determined a density of 1227.27 kg/m$^3$ for date-palm fibres, considering it to be lighter compared to cotton, jute, flax or sisal, or around 1200 kg/m$^3$ for date-palm and date-palm/bamboo fibre-reinforced epoxy hybrid composites [28]. So, one of the advantages brought by the proposed laminate composites in this study is their light weight compared with other potential similar structures reinforced with natural or synthetic fibres, such as Kevlar, carbon and glass epoxy composites, for which the densities are greater than 1230 kg/m$^3$, as was found in the literature [25].

### 3.2. Microscopic Investigation

The microscopy of the date-palm fibres, and of the *Calotropis* fibres (both untreated and treated) with 240× magnification, are shown in Figure 5a–c.

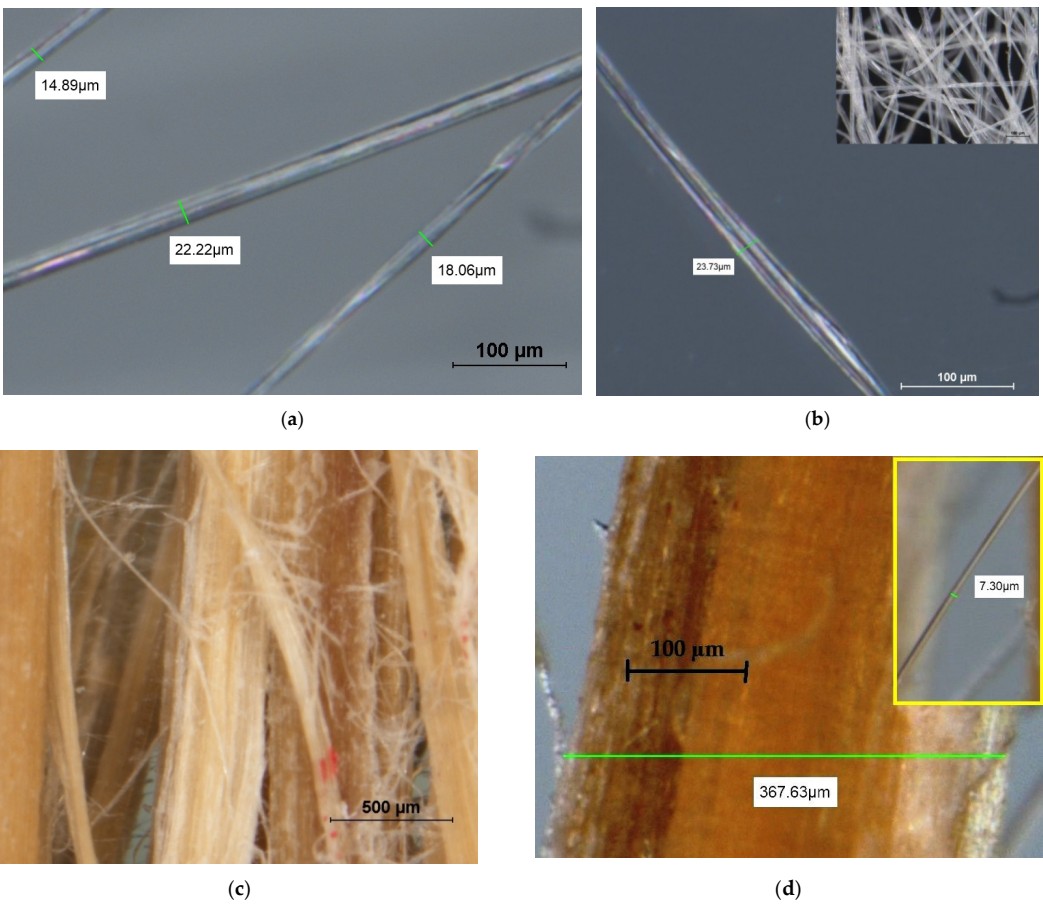

**Figure 5.** Microscopic investigation of the fibres: (**a**) untreated *Calotropis* fibres (240× magnification); (**b**) treated *Calotropis* fibre (240× magnification); (**c**) date-palm fibres (60× magnification); (**d**) measured bundle of date-palm fibres and the measured diameter of the single fibre in the detail image (240× magnification).

The measurements of the fibres' diameters reached values between 0.22 mm and 0.37 mm for the date palm, between 0.015 mm and 0.022 mm for the untreated *Calotropis*, and between 0.023 mm and 0.031 mm for the treated *Calotropis*. The measurements of untreated *Calotropis* are in line with the measurements recorded by other researchers based on SEM micrographs [20]. They also noticed that treating *Calotropis* with NaOH

for 5 min at room temperature produces winding fibres. Other researchers observed that the treatment with NaOH significantly increased the diameter of the fibres [22], a fact revealed by microscopic investigation from the present research. As a first conclusion to the microscopic investigation, the *Calotropis* fibres have diameter sizes ten times smaller than those of the date-palm fibres. As seen in Figure 5a,b, the *Calotropis* fibre is transparent, and no visible transformation was noticed when comparing the treated one with the untreated one, except the increased diameter.

As regards the date-palm fibres, as they appear in Figure 5c,d, they are cylindrical bundles composed of single fibres measuring less than 10 μm in diameter.

The microscopic images of D, DC, and DTC are presented in Figure 6. The images shown in Figure 6a–c were taken with 22.5× magnification, and the detail presented in Figure 6d was 60× magnified. The images were taken on the edge of the composite specimen, so that the three-layer in the case of the D composite and the five-layer composites for both DC and DTC were visible.

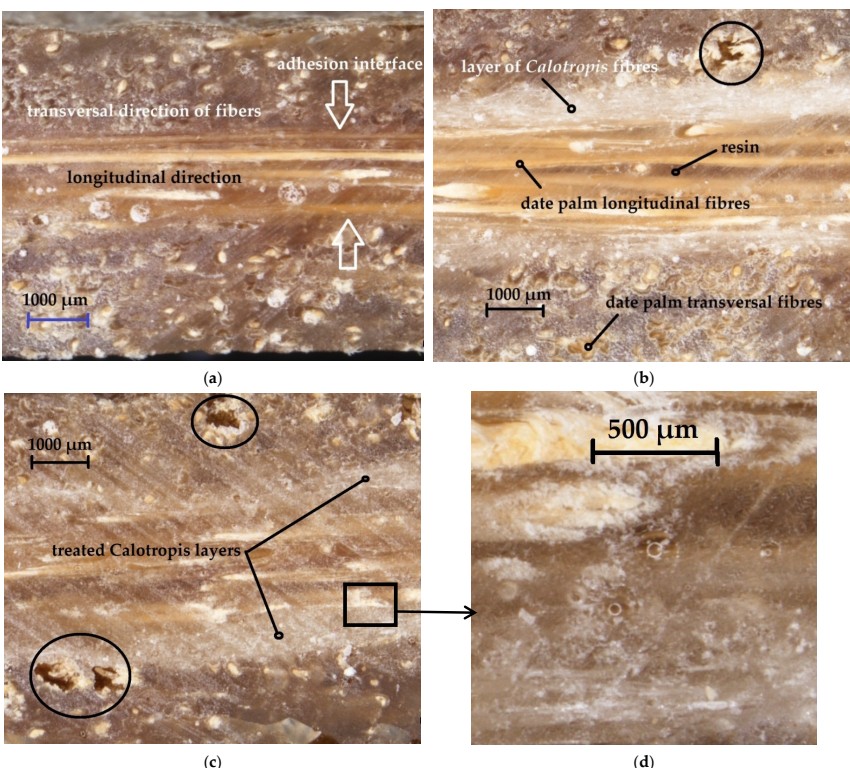

**Figure 6.** Microscopic investigation of the composite structures: (**a**) D (22.5× magnification); (**b**) DC (22.5× magnification); (**c**) DTC (22.5× magnification); (**d**) detail of the interface between date-palm layer and treated *Calotropis* layer (60× magnification) marked at (**c**) with black rectangle.

All three structures were placed under the microscope so as to point out the longitudinal section of the date-palm fibres for the core and their crosscut section for the faces.

In Figure 6a, representing the edge of the three-layer composite (D), the compact structure and the good adhesion interface between the layers can be observed. For the second composite (Figure 6b) made from three layers of date-palm fibres and two layers of untreated *Calotropis* sheets, the longitudinal and crosscut sections of the date-palm fibres are highlighted. The dark brown color from this image represents the hardened epoxy resin. The layer of untreated *Calotropis* fibres is white colored and very well visible in Figure 6b. Less visible is the layer of treated *Calotropis* fibres in Figure 6c, corresponding to the composite DTC. An explanation could be the yellowish color of this layer due to the chemical treatment applied, but also a possible better impregnation with epoxy resin. The circled areas highlighted in Figure 6a,b represent the gaps (voids) that occurred in the

composite structures, especially in the vicinity of the *Calotropis* layers. They are caused by the insufficient impregnation with resin in certain areas, as a result of the hand layup technology used for impregnation, which could not ensure uniformity on the whole surface.

The existence of the voids at the interface area located between date-palm fibres and epoxy resin matrix was unveiled also by other researchers in their study [29]. In comparison, bamboo fibres displayed a better consolidation in the matrix resin and a greater contribution to the mechanical strength. Mixing date-palm fibres with bamboo fibres in an epoxy resin matrix could be the subject of further research, having as an objective the improvement of the mechanical performance of such composite materials.

### 3.3. FTIR Analysis

For the untreated *Calotropis* fibre (Control), the spectra (Figure 7) presented high absorptions bands at around 3330 $cm^{-1}$, 2910–2800 $cm^{-1}$, 1730 $cm^{-1}$, and several peaks in the fingerprint region (presented in Figure 7), especially in the ranges 1640–1500 $cm^{-1}$ and 1370–900 $cm^{-1}$, indicating the presence of the main constituents of lignocellulosic natural materials such as cellulose, hemicelluloses, and lignin, in good accordance with the literature [39]. The high and broad absorption with a maximum at around 3330 $cm^{-1}$ is assignable to H-bonded hydroxyl (-OH) groups in alcohols (cellulose, hemicelluloses) and phenols (lignin), while absorption at 2910 $cm^{-1}$ is assigned to C-H stretching vibrations in methylene (-$CH_2$-) and methyl (-$CH_3$) groups in cellulose, hemicelluloses, and also possibly in waxes or oils, if present. The high absorption at 1730 $cm^{-1}$ (unconjugated carbonyl) can be mainly associated with the acetyl groups in hemicelluloses and possibly also with other esters, such as waxes or oils, which might be present on the fibres [40]. Another absorption band is that at 1604 $cm^{-1}$ (aromatic ring) with a shoulder at about 1645 $cm^{-1}$, which might be assigned to conjugated carbonyl bonds or aromatic ketones in the structure of lignin, but also to O-H bending vibration for absorbed water [39,41]. The absorption at 1370 $cm^{-1}$ is assigned to cellulose and hemicelluloses, while that at 898 $cm^{-1}$ (C-H vibration) is characteristic of crystalline cellulose. The strong absorption at 1240 $cm^{-1}$ might be assigned to the acetyl groups in hemicelluloses: 1255 $cm^{-1}$ according to [41], 1230 $cm^{-1}$ assignable to -C-C plus C-O plus C=O stretch for acetyl in xylan according to [41], and also to the syringyl ring in the structure of lignin, 1235 $cm^{-1}$ syringyl ring in lignin, and C-O stretch in lignin and xylan, according to [42,43]. The absorption at 1507 $cm^{-1}$ is the most characteristic absorption of lignin (aromatic skeletal vibration).

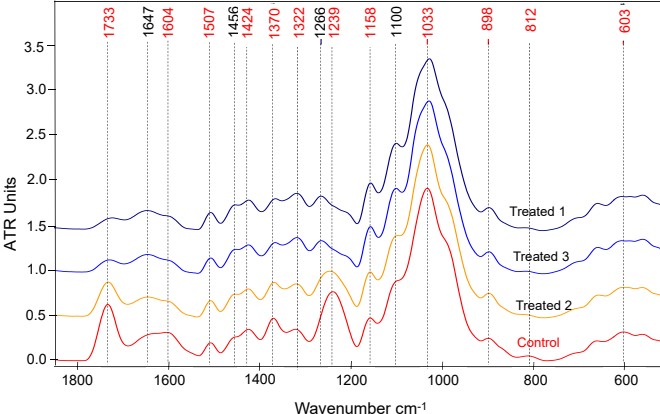

**Figure 7.** FTIR analysis of the treated and untreated *Calotropis* fibres (comparative spectra in fingerprint region 1800 $cm^{-1}$–600 $cm^{-1}$).

For the treated *Calotropis* fibre (sample 2), the spectrum was quite similar to the control sample, with some differences in the intensity of some characteristic absorptions, as follows: slight increase of -OH absorption band at 3300 $cm^{-1}$; a slight decrease of absorptions at 1733 $cm^{-1}$ and 1370 $cm^{-1}$, corresponding to the decrease of hemicelluloses content; the increase of absorption at 1645 $cm^{-1}$, which becomes evident whilst the former

absorption peak at 1604 cm$^{-1}$ decreased to a shoulder, possibly indicating some changes in the structure of aromatic compounds (lignin) and/or increased content of absorbed water due to an increased hygroscopicity. The absorption at 1239 cm$^{-1}$ decreased similarly to the band at 1730 cm$^{-1}$, most likely pointing out degradation of hemicelluloses, or saponification (hydrolysis in an alkaline medium) with elimination of the acetyl groups.

The spectra for the treated samples 1 and 3 were similar to one another while both showing more substantial changes when compared to the spectrum for control *Calotropis* fibres than was the case for treated sample 2. The main changes were: increase of the -OH groups absorption band at 3300 cm$^{-1}$; more significant decrease of absorption at 1733 cm$^{-1}$; clear evidence of the absorption at 1645 cm$^{-1}$, which includes as a shoulder the absorption at 1604 cm$^{-1}$; the disappearance of the absorption peak at 1239 cm$^{-1}$ or its shifting to 1265 cm$^{-1}$, assignable to guaiacyl ring plus C=O stretch in guaiacyl lignin [40]. This might indicate two types of processes: (1) decrease of acetyl groups in xylan (hemicelluloses) by alkaline hydrolysis or decrease of hemicelluloses content (correlated with the decrease of 1733 cm$^{-1}$) and (2) de-metoxylation of lignin, respectively transforming some syringil rings in the structure of lignin into guayacil rings. Additionally, the small absorption peak at 899 cm$^{-1}$ seems to be slightly increased, indicating a relative increase of (crystalline) cellulose content due to degradation and removal of other components, mainly hemicelluloses, as the most characteristic lignin absorption band at 1507 cm$^{-1}$ seems to be little affected by the applied treatment.

FTIR spectra indicated that the alkaline treatment applied in this research brought about a de-acetylation and decrease of hemicelluloses content by saponification and alkaline hydrolysis and possibly some chemical changes in the structure of lignin, such as some demethoxylation of the syringil rings. Accordingly, the FTIR spectra highlighted an apparent increase in cellulose content and free hydroxyl groups. These findings are in good accordance with similar reported research. For instance, in [40] it was found that the pre-treatment of *Calotropis* fibres with sodium hydroxide resulted in an increase of cellulose content from 64.47% to 69.93% and its crystallinity index from 36% to 39.8%, while hemicelluloses content decreased from 9.64% to 6.72%, lignin decreased from 13.56% to 11.25% and wax content also decreased from 1.93% to 1.12%. All these changes had a positive influence on the characteristics of the selected fibres, which are important for their utilization as reinforcing materials in polymer eco-friendly composites.

### 3.4. Water Immersion

The recorded values for the water absorption (WA) and thickness swelling (TS) are presented in Figure 8 for the D and DC composites investigated.

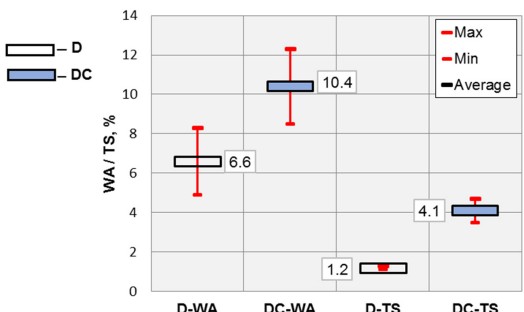

**Figure 8.** Water absorption (WA) and thickness swelling (TS) after 24 h of immersion of D and DC composites into water.

The presence of the *Calotropis* fibres in the structure of the composite reduces the water resistance, increasing the values recorded for water absorption (WA) and thickness swelling (TS) in the conditions of immersing the samples in water for 24 h. As seen in Figure 8, the recorded average value of WA for the three-layer composite made of date-palm fibres (D) was 6.6% and higher with 57.6%, up to a value of 10.4% for the composite with two



additional layers of untreated *Calotropis* fibres (DC). The thickness swelling (TS) increased for composite DC with 3.42%. The higher recorded values of WA and TS can be explained by the hydrophilic character of treated and untreated *Calotropis* fibres, highlighted by the high content of hydroxyl groups that increased following alkaline treatment, revealed by FTIR in this research and other studies [17,22]. One of these studies [22] showed that the natural wax coating of these fibres is partially removed when an alkaline treatment is applied, thus improving their absorption property. This property is in favor of its applicability as an absorbent but becomes a disadvantage when considering the water behavior of *Calotropis* fibre-reinforced composites. Therefore, for the potential applications of the composites developed in this research in humid environments, or in water contact (for boats for example), the use of *Calotropis* fibres in the structure is not recommended. The performance of *Calotropis procera* fibres as reinforcements in an epoxy matrix was investigated by several researchers [44] using fibres extracted from the branch of the tree. For 30 weight % fibres in the composite structure, the WA value was 7 %, lower than the results obtained for the five-layer laminates proposed in this study. Date palm/bamboo fibre-reinforced epoxy composites immersed in water for eight days [29] showed a fast increase in the first 24 h, with the TS and WA recorded values around 6%, similar to structure D investigated in this paper. Another study [2] showed that the date-palm fibres have a lower porous structure and implicitly a lower absorption capacity compared with sisal, wheat straw, hemp and kenaf fibres, providing the important advantage of using date palm in the place of other natural fibres for composite manufacturing.

### 3.5. Mechanical Performance

The results recorded for modulus of elasticity, both for tensile and bending tests, tensile strength, flexural strength and impact strength are presented in Figure 9.

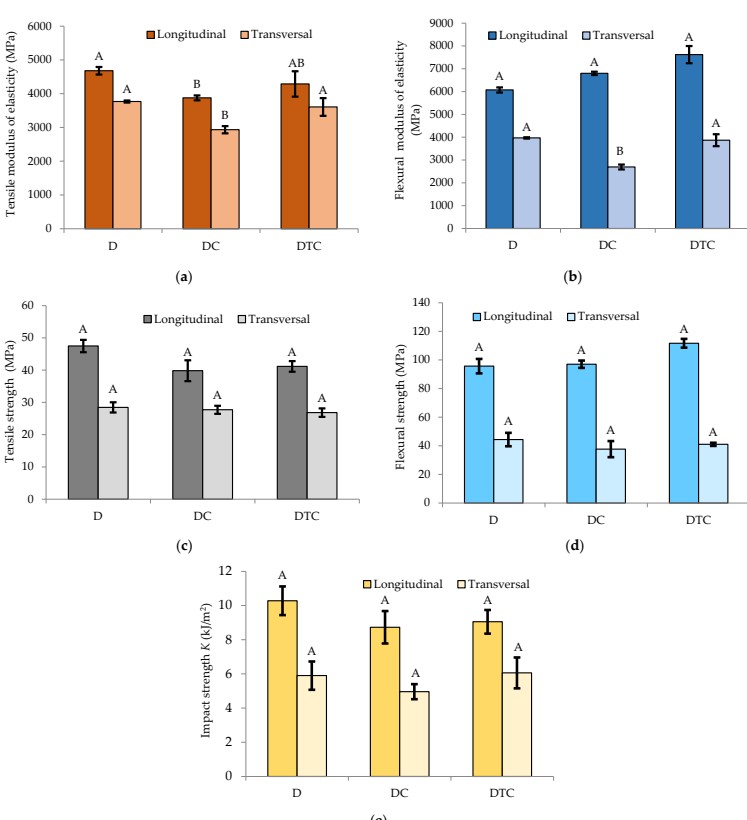

**Figure 9.** Comparison of the mechanical properties for the tested composite materials (variables with the same letter mean that the difference is not statistically significant): (**a**) tensile modulus of elasticity; (**b**) flexural modulus of elasticity; (**c**) tensile strength; (**d**) flexural strength; (**e**) impact strength.



For the longitudinal tested samples, the highest average values of tensile strength and tensile modulus of elasticity, respectively, were recorded for the three-layer structures D (47.5 MPa and 4678.6 Mpa, respectively), followed by five-layer structures with treated *Calotropis* DTC (41.2 MPa and 4289 Mpa, respectively) and untreated DC (39.8 MPa and 3877 Mpa, respectively). The same trend was noticed for impact strength (10.3 MPa for D composite, 9.1 MPa for DTC and 8.7 MPa for DC). Instead, the flexural modulus of elasticity and flexural strength values increased with the addition of *Calotropis* fibres, the highest ones being reached by DTC composites with treated *Calotropis* fibres (average values of 7623 MPa for flexural modulus of elasticity and 111.7 MPa for flexural strength). Average values recorded for DC/D structures were as follows: 6802 MPa/6072.5 MPa for the flexural modulus of elasticity and 97.1 MPa/95.7 for the flexural strength.

The mechanical strength values recorded for the three-layer laminate (D) tested on the longitudinal direction are higher (flexural strength was 1.5 times higher and the tensile strength was 1.2 times higher) than those recorded in another study [28] for panels manufactured with 50:50 as weight ratio of date-palm fibres to epoxy resin, using date-palm fibres from different parts of the plant. As regarding the flexural modulus of elasticity, with the better interfacial adhesion and dispersion of fibres and matrix, there are better flexural modulus results [29]. As found in the literature [28,30–32], alkali treatment applied to natural fibres improves the adhesion between the fibres and matrix, improving the mechanical performance of the resulted composite. The results obtained in this research confirm that the chemical treatment applied to the *Calotropis* fibres led to the increase of the mechanical characteristics of the composite material DTC compared with the ones of the composite DC. In Figure 9a, it is shown that the chemical treatment of the *Calotropis* fibres led to a tensile modulus of elasticity 10.66% and 22.95% higher than the one corresponding to the composite DC containing untreated *Calotropis* fibres in longitudinal and transversal directions of the fibres, respectively. Figure 9b,d show that the chemical treatment of the *Calotropis* fibres has increased the flexural modulus of elasticity by 12.1% and the flexural strength by 15% for the DTC composite compared to DC composites.

The flexural modulus of elasticity in the longitudinal direction of the fibres for DC and DTC composites is higher by 12.02% by 25.54%, respectively, than the flexural modulus of elasticity of composite D (Figure 9b). Contrarily, the flexural modulus of elasticity in the transversal direction of the fibres decreased by 32.07% or 2.52% by adding the two layers of untreated or treated *Calotropis* fibres, respectively, compared to the flexural modulus of elasticity obtained for composite D (Figure 9b).

The alkali treatment of *Calotropis* fibres proved to be beneficial also for the tensile modulus of elasticity and tensile strength (Figure 9a,c), but the increasing percentages turned out to be lower. Impact strength is included in the same category (Figure 9e). The impact strength of the composite material DTC improved by less than 5% compared to the composite DC when untreated *Calotropis* fibre sheets were replaced by the treated ones (Figure 9e).

Epoxy composites reinforced with 10 up to 40 wt% of the *Calotropis* fibres [44] recorded flexural strength values between 25 MPa and 29 MPa, tensile strength around 11 MPa and impact strength between 1 kJ/m$^2$ and 1.5 kJ/m$^2$, values approximately four times lower than those recorded for epoxy composites reinforced with date-palm fibres (D composite). Based on these results, the decrease of the mechanical strength of the laminates with the participation of *Calotropis* fibres (DC structures) can be explained by their low bearing capacity.

The flexural strength on longitudinal direction of the reinforcement fibres was 1.46% or 16.72% higher for the composite containing untreated or treated *Calotropis* fibres, respectively, than the one obtained for the composite material D, which did not contain *Calotropis* fibres, as shown in Figure 9d. Contrarily, the flexural strength evaluated in the transversal direction of the reinforcement fibres recorded a small decrease by adding either the untreated or treated *Calotropis* fibres, as shown in Figure 9d.

The impact strength in the longitudinal direction of the reinforcement fibres was 15.53% or 11.65% lower for the composite materials containing untreated or treated *Calotropis* fibres, respectively, than the impact strength determined for the composite material D, which did not contain *Calotropis* fibres, as shown in Figure 9e. This behavior was expected as long as the flexural modulus of elasticity had increased by adding fibres (Figure 9b) and the elastic failure energy stored by the impact was inversely proportional to the flexural modulus of elasticity. Just a small variation in the impact strength in the transversal direction of the reinforcement fibres was observed for the composites containing *Calotropis* fibres—composites DC and DTC—with respect to the impact strength determined for the composite material D, which did not contain *Calotropis* fibres.

The resulting values for tensile strength in the present research are similar to those of woven kenaf fibre-reinforced acrylonitrile-butadiene-styrene laminates [28]. As this study shows, the orientation of fibres in the structure influences the mechanical performance of the composite material. The results presented in Figure 9 show that having the orientation of the samples in the transversal direction when conducting the mechanical tests results in approximately half of the mechanical strength recorded for the longitudinal orientation of the samples for all tests.

As the SEM analysis performed by some researchers showed, the *Calotropis* fibres treated with NaOH for 5 min at room temperature tend to stick together [20], which can explain why the agglomeration of the fibres in some areas is to the detriment of the others (as seen in Figure 2d), thus affecting the strength of the composites in some areas and recording a larger field of data scattering. Fibres extracted from the stem of the *Calotropis* tree used as reinforcement in an epoxy composite [23] showed higher values of tensile strength when an alkali treatment was applied ($46.21 \text{ N/mm}^2$ compared to $39.01 \text{ N/mm}^2$). An increase from 39.84 MPa to 41.17 MPa has been also recorded in the present research for structure DTC compared to structure DC, showing that the treatment applied to *Calotropis* fibres was beneficial to the tensile strength results when the samples were tested in the longitudinal direction. The flexural strength and impact strength had the same improvement trend for structure DTC compared to DC, a trend also noticed by [23] for alkali-treated fibres. The improvement of the mechanical properties of composite DTC compared to composite DC is attributed to a better impregnation of the treated *Calotropis* fibres compared to the untreated ones, an aspect revealed by the observations made through the microscopic analysis of these composites.

Natural fibres with higher mechanical strengths have higher cellulose content [30]. According to the FTIR analysis performed for treated and untreated *Calotropis* fibres, an apparent increase of cellulose content and free hydroxyl groups was recorded for the treated fibres, thus explaining the higher values recorded for tensile and bending strength and also for impact strength in the case of composites containing *Calotropis* treated fibres. From the values recorded for mechanical strengths, the influence of the direction of testing is evident, with lower values being recorded for the transversal direction in all cases and for all composites.

The existence of the natural waxy substance on the surface of date-palm fibre does not allow a strong bond with epoxy polymer matrix. Having the advantage of being lighter than other natural fibres, chemically modified date-palm long fibres can be used as a reinforcing component to improve the tensile and flexure strength in fibre-reinforced biocomposites, and these may be also used as substitutes for heavy-weight civil engineering materials [31]. Further research can be performed to improve the bond at the interface area between the date-palm fibres and epoxy polymer matrix.

### 3.6. Statistical Analysis

The addition of the treated and untreated *Calotropis* fibres into the structure of the composite materials has significantly ($p < 0.05$) affected both the tensile and flexural modulus of elasticity in the case of tests applied in the transversal direction of the fibres. Additionally, both impact strength and tensile strength were not statistically significantly affected by the

applied method of testing (longitudinal or transversal) in the presence of *Calotropis* fibres in the composite structure. For the tensile modulus of elasticity in the longitudinal direction of fibres, the treatment of *Calotropis* fibres had a significant effect ($p = 0.03$). Additionally, the presence of *Calotropis* fibres in the structure DC showed a statistical significantly effect on the thickness swelling (TS) and water absorption (WA). The grouping of the statistical classes for each test's method are clearly presented separately in the comparative diagrams in Figures 4d and 9a–e. The variables with the same letter mean that the difference is not statistically significant.

## 4. Conclusions

This study showed that long date-palm fibres extracted from midribs are attractive natural resources to be used for structural composites based on laminar epoxy resin as binder.

The investigation of the mechanical properties of the *Phoenix dactylifera/Calotropis procera* fibre-reinforced epoxy hybrid composites showed that the presence of *Calotropis* fibres (treated and untreated) in five-layer composite structures (DTC and DC) resulted in lower values of tensile/impact strength compared to those of the three-layer composites made only from date-palm sheets (D). The same trend was noticed for the tensile modulus of elasticity. Instead, the flexural strength and flexural modulus of elasticity were improved with the contribution of *Calotropis* fibres. An increase in water absorption (WA) and thickness swelling (TS) values was recorded in the presence of *Calotropis* fibres in the five-layer composites.

The alkali treatment of the *Calotropis* fibres improved the tensile and impact strength of the composites compared to the ones made with untreated fibres, but not enough to reach the tensile and impact strength values recorded by the three-layer structures made only from date-palm fibres. The increase in strength in this case is explained by an apparent increase of cellulose content and free hydroxyl groups as result of the decrease in hemicellulose content caused by saponification and alkaline hydrolysis, while some chemical changes in the structure of lignin also seem possible.

The transversal direction of testing the samples resulted in the decreasing of the mechanical strength of the composites by about half of that recorded for the longitudinal direction of the fibres, in all mechanical tests (tensile test, bending test and Charpy impact test) involved in this research.

It must be mentioned that just the flexural modulus of elasticity and flexural strength are greater for composites DC and DTC, which contain *Calotropis* fibres, than for composite D, which does not contain *Calotropis* fibres. Moreover, it may be remarked that the chemical treatment of the *Calotropis* fibres led to the flexural modulus of elasticity being 12% and 43.5% higher than the one corresponding to the composite DC containing untreated *Calotropis* fibres in longitudinal and transversal directions, respectively. The increase of the flexural modulus of elasticity caused by adding treated or untreated *Calotropis* fibre sheets in composites is the cause of the decrease of the impact strength, which varies inversely proportionally to the flexural modulus of elasticity.

The potential applications of the composites developed might include interior components in the automotive industry or maritime industry, but further research is necessary.

**Author Contributions:** Conceptualization, H.Z.H., C.C. (Camelia Coșereanu) and M.H.M.; methodology, H.Z.H., C.C. (Camelia Coșereanu) and C.C. (Camelia Cerbu); software, M.C.T.; validation, H.Z.H., C.C. (Camelia Coșereanu) and C.C. (Camelia Cerbu); formal analysis, M.H.M. and H.Z.H.; investigation, M.C.T., S.V.G. and C.C. (Camelia Cerbu); resources, M.H.M. and C.C. (Camelia Coșereanu); data curation, M.C.T., C.C. (Camelia Cerbu), M.H.M. and H.Z.H.; writing—original draft preparation, C.C. (Camelia Coșereanu) and C.C. (Camelia Cerbu); writing—review and editing, M.C.T., C.C. (Camelia Cerbu) and H.Z.H.; visualization, M.H.M., S.V.G. and H.Z.H.; supervision, H.Z.H., C.C. (Camelia Coșereanu) and C.C. (Camelia Cerbu); project administration, C.C. (Camelia Coșereanu); funding acquisition, H.Z.H. and C.C. (Camelia Coșereanu). All authors have read and agreed to the published version of the manuscript.

**Funding:** This research received external funding from Armaghan e Tabiat e Makkoran Co. (National Identification No. 14007930167, Registration No. 1658, and Economic code 411638437544) by contract No. 99-7424475.

**Acknowledgments:** We hereby acknowledge the structural funds project PRO-DD (POS-CCE, O.2.2.1., ID 123, SMIS 2637, No. 11/2009) for providing the infrastructure used in this work (https://icdt.unitbv.ro/en/research-and-development-projects/the-r-d-institute-project.html, accessed on 7 November 2022).

**Conflicts of Interest:** The authors declare no conflict of interest.

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
