# Peer review of "Investigation on Phoenix dactylifera/Calotropis procera Fibre-Reinforced Epoxy Hybrid Composites"

_forests, doi:10.3390/f13122098_

Round 1

Reviewer 1 Report

This article describes the process of making a composite from date palm fibers and Calotropis procera. The important question is: does date palm (P dactylifera) considered as forest tree? this is crucial point to accept this paper in forest journal. However,  I have some comments:
Introduction: According to the title, it is necessary to describe the importance of the date palm then the other  used specie.
MM: section 2.1.1. Please give a description of the role of each solution used (water, NaOH, etc.). WHY do you use midrib? are there other date palm materials that can be used? like bunches?
RD: well done but added statistical classes in the graphs. Add photo of the final ptoduct. 

Author Response

Answers to Reviewer 1

This article describes the process of making a composite from date palm fibers and Calotropis procera. The important question is: does date palm (P dactylifera) considered as forest tree? this is crucial point to accept this paper in forest journal. However,  I have some comments:

Answer:.

No, It is not a forest. Regarding most of the date palm trees being hand planted, it considers a Grove as mentioned on page 4, line 118.

The world’s total number of date palms is more than 120 million (Based on the below link).

https://doi.org/10.1016/j.jclepro.2020.120951

Approximately 11.8 million hectares are under cultivation of date palm trees, distributed in 94 countries. Overall, the date palm waste, generated from seasonal pruning and trimming can be estimated at an average of 35 kg per tree (Based on the below link).

https://doi.org/10.3390/fib10040035

Considering the 120 million date palm trees, and 35 kg of waste per tree, 4,200,000 tons of natural fibers is existed to consume in different various fields. Although, date palms were used for 360 purposes in life (Based on https://doi.org/10.3390/fib10040035). Consequently, using the waste of annual pruning date palm trees as natural fiber helps prevent cutting forest trees around the world.

Iran is the third and second country under cultivation of date palm trees in the world and the middle east, respectively. Almost, 21% of all groves in the world belong to Iran. (Based on the Date palm fiber composites processing, properties, and applications book, the Date palm cultivation book, and the statistics of the Food and Agriculture Organization of the United Nations (FAO)).

Furthermore, Iran has 214,000 hectares under cultivation of date palm (Based on the below link).

https://doi.org/10.17660/ActaHortic.2004.655.40

Introduction: According to the title, it is necessary to describe the importance of the date palm then the other used specie.

Answer:

The importance of date palm was introduced in the text, as follows: Considering the 120 million date palm trees, and 35 kg of waste per tree, 4,200,000 tons of natural fibers is existed to consume in different various fields. Although, date palms were used for 360 purposes in life (Based on https://doi.org/10.3390/fib10040035). Consequently, using the waste of annual pruning date palm trees as natural fiber helps prevent cutting forest trees around the world.

MM: section 2.1.1. Please give a description of the role of each solution used (water, NaOH, etc.).

Answer:

The NaOH solution was used for delignification. The acetic acid solution was consumed for neutralization of the alkaline state. However, the washing steps with water were done to cleanliness and further delignification [21] (https://doi.org/10.1016/j.indcrop.2020.112466). The explanations were added in the subchapter 2.1.1. Date palm midrib long fibre extraction

WHY do you use midrib? are there other date palm materials that can be used? like bunches?

Answer:.

It is not possible to extract long fiber from other parts of the date palm tree, except for Midrib (Rachis). Regarding the anatomic features of date palm parts, just the midrib consisted of long fibers posited in the natural matrix (Based on the below link).

https://doi.org/10.1007/s10570-019-02259-6

The explanation was added in the text at the begining of the subchapter 2.1.1. Date palm midrib long fibre extraction

RD: well done but added statistical classes in the graphs. Add a photo of the final product. 

Answer:

The statistical classes were added to the graphs. Furthermore, the photo of the final composite was placed in Figure 3e..

The authors thank you for your valuable comments, which helped us to improve the scientific level of the manuscript.

Reviewer 2 Report

Dear authors.

The paper has been revised. The paper brings the elaboration of composites with epoxy resin and fibers. The work presents good organization, good methodology and reproducibility. Some suggestions:

- Figure 3 shows bad resolution. Improve the resolution and increase the font. Note that the fonts are different between figures.

- In figure 5d, there is no need for the black border. Maintain standardization.

- In figure 6, the font is too small. Is it necessary to use the chart provided by the equipment?

- Why was there no statistical analysis between treatments? Only then would it be possible to indicate specific variations considering the standard deviation of the treatments. At the end of the paper I found a statistical analysis section. This analysis, with average tests, should be in all results figures.

Author Response

Answers to Reviewer 2

Dear authors.

The paper has been revised. The paper brings the elaboration of composites with epoxy resin and fibers. The work presents good organization, good methodology and reproducibility. Some suggestions:

- Figure 3 shows bad resolution. Improve the resolution and increase the font. Note that the fonts are different between figures.

Answer: Figure 3 became Fig. 4. The resolution was improved and the font was changed.

- In figure 5d, there is no need for the black border. Maintain standardization.

Answer: The black border is now deleted.

- In figure 6, the font is too small. Is it necessary to use the chart provided by the equipment?

Answer: The font was changed. The chart provided by the equipment shows the chemical changes and their positive influence on the fibers characteristics, which are important for their utilization as reinforcing materials in polymer composites.

- Why was there no statistical analysis between treatments? Only then would it be possible to indicate specific variations considering the standard deviation of the treatments. At the end of the paper I found a statistical analysis section. This analysis, with average tests, should be in all results figures.

Answer: It was done. The statistical analysis has been improved and it is comprised in the figures. Also, additional explanations are now in the text.

The authors thank you for your valuable comments, which helped us to improve the scientific level of the manuscript.

Reviewer 3 Report

The writing style of the manuscript should be thoroughly modified and resubmitted.

Abstract is lengthy. It should be written short and crisp.

Introduction lacks enough motivation.

What is the significance and application of the present study?

More number of references are clubbed. Please discuss the pros and cons of individual work.

Results and discussion lacks scientific rigour. It reads like a experimental report.

What do you mean by MOE/MOR? Please avoid abbreviations.

Figure 6 is not readable. Please increase the font size.

Conclusion should be written point wise highlighting the research results.

Author Response

Answer to Reviewer 3

The writing style of the manuscript should be thoroughly modified and resubmitted.

Answer: Modifications of the writing style were done throughout the manuscript.

Abstract is lengthy. It should be written short and crisp.

Answer: The abstract was shortened and it is more concise.

Introduction lacks enough motivation.

Answer: The introduction has been restructured and the information has been organized in such a way as to emphasize the importance of this study.

What is the significance and application of the present study?

Answer: The significance and application of the present study is emphasized at the end of the modified Introduction chapter, as follows:

The present paper proposes three structures of epoxy-resin based composites using as raw materials. In the present study, three different laminate were developed, by reinforcing long date palm (Phoenix Dactylifera L.) fibres extracted from the midribs and Calotropis white silk extracted from the mature fruits of the tree in epoxy resin, in order to obtain composites to be used in applications such as the automotive, aerospace and construction industries, able to substitute the synthetic fibres based composites, which are more expensive and require complex manufactured technologies. The advantage of using these natural fibres is the availability of a large amount of waste, their low cost and low CO2 footprint. The research conducted on the developed composites investigates the effect of factors like reinforcement type, alkali treatment of Calotropis silk, and number of ply in the laminate on the mechanical behavior of the composites. For a more comprehensive study, a stereo-microscopic analysis of the fibres and of the adhesion between layers, investigation of the vertical density profile (VDP), FTIR analysis and resistance to water are supporting the interpretation of the mechanical results, helping in recommending The fibres formed in the shape of flat sheets are intended to be used in three-layer composites (made from date palm sheets with adjacent layers having perpendicularly oriented fibres) and two sets of five-layer composites (with two additional layers of Calotropis sheets, untreated for one set and alkali treated for the other set) bonded by epoxy resin. Microscopy of the fibres and of the composites, vertical density profile (VDP), FTIR analysis, and mechanical testing under tensile, bending, and impact loads completed by the immersion test into the water are to be carried out in order to characterize the physical and mechanical properties of the proposed composites, which could recommend these materials in indoor or outdoor applications, such as automotive, construction industry or ship hulls.

More number of references are clubbed. Please discuss the pros and cons of individual work.

Answer: The discussion chapter was considerably improved and pros and cons arguments of the conducted study were added.

Results and discussion lacks scientific rigour. It reads like a experimental report.

Answer: Results and discussion chapter was modified and the manner of exposing the results was improved for a higher scientific level.

What do you mean by MOE/MOR? Please avoid abbreviations.

Answer: The abbreviations were removed.

Figure 6 is not readable. Please increase the font size.

Answer: The figure was modified and changed.

Conclusion should be written point wise highlighting the research results.

Answer: Conclusion chapter was modified as suggested by the reviewer.

The authors thank you for your valuable comments, which helped us to improve the scientific level of the manuscript.

Round 2

Reviewer 1 Report

All necessary corrections have been taken into account by the authors, I recommend the publication of this article after minor revision:

Please add this statement in the title of Figure 9: Variables with the same letter mean that the difference is not statistically significant.

Author Response

Thank you for your recommendation. The statement was added in the title of the Figure 9.

The authors thank you for your effort in reviewing our manuscript. 

Reviewer 2 Report

Dear All,

The authors have improved the presentation of the paper. I believe that the paper can be accepted for publication.

Author Response

The authors thank you for all your remarks and for helping us to improve the quality of the manuscript.

Reviewer 3 Report

The Authors have carried out revision of the manuscript. However, still there is scope for improvement.

Authors can get help from a native speaker and use correct terminology appropriately.

Scientific depth is still missing in the results and discussion.

Please change the notation for kilograms as kg and avoid Kg in all the sections of the manuscript.

In conclusion, please use the values of the results obtained in the present study to draw meaningful concluding remarks.

Title is not reflecting the contents of the manuscript.

The suggested title may be Investigation on Phoenix Dactylifera/Calotropis procera fibre reinforced epoxy hybrid composites

Author Response

Answers to Reviewer 3

Question 1: The Authors have carried out revision of the manuscript. However, still there is scope for improvement.

Answer 1: The authors would like to thank to Reviewer 3 for the appreciation regarding the semnificative improving of the manuscript after the first review step.

Considering the Reviewer 3’s suggestion, the authors continue to improve their manuscript and the second revised version of our manuscript contains additional text regarding the interpretation and discussions on the results.

Question 2: Authors can get help from a native speaker and use correct terminology appropriately.

Answer 2: The Reviewer 3 has right and as a result, the revised version of the manuscript contains corrections regarding the terminology and expression corrections.

Question 3: Scientific depth is still missing in the results and discussion.

Answer 3: The authors would like to mention that the mechanical properties of the composite materials investigated are already explained by considering the results obtained by both FTIR analysis and microscopic investigations. The authors improved the discussions of the results by adding three new paragraphs in section “3.5. Mechanical performance” in order to make comments on the results obtained in their research. The new paragraphs added in the revised version of the manuscript are shown below.

“The results obtained in this research confirm that the chemical treatment applied to the Calotropis fibres led to the increase of the mechanical characteristics of the composite material DTC compared with the ones of the composite DC. In Figure 9(a) it is shown that the chemical treatment of the Calotropis fibres led to tensile modulus of elasticity with 10.66 % and 22.95 % higher than the one corresponding to the composite DC containing untreated Calotropis fibres in longitudinal and transversal directions of the fibres, respectively.”

“The flexural modulus of elasticity on longitudinal direction of the fibres for DC and DTC composites is higher with 12.02 % or with 25.54 % respectively, than the flexural modulus of elasticity of the composite D (Figure 9b). Contrary, the flexural modulus of elasticity on transversal direction of the fibres, has decreased with 32.07 % or with 2.52 % by adding the two layers of untreated or treated Calotropis fibres, respectively, compared to the flexural modulus of elasticity obtained for the composite D (Figure 9b).”

“The flexural strength on longitudinal direction of the reinforcement fibres was 1.46 % or 16.72 % higher for the composite containing untreated or treated Calotropis fibres respectively, than the one obtained for the composite material D which did not contain Calotropis fibres, as shown in Figure 9(d). Contrary, the flexural strength evaluated in transversal direction of the reinforcement fibres has recorded a little decreasing by adding either of the untreated or treated Calotropis fibres as shown in Figure 9(d).”

“The improving of the mechanical properties of the composites DTC compared to the composites DC, is attributed to a better impregnation of the treated Calotropis fibres compared to the untreated ones, an aspect revealed by the observations made through the microscopic analysis of these composites.”

Question 4: Please change the notation for kilograms as kg and avoid Kg in all the sections of the manuscript.

Answer 4: The notation for kilograms was changed in kg in all sections of the manuscript and in Fig. 4(d).

Question 5: In conclusion, please use the values of the results obtained in the present study to draw meaningful concluding remarks.

Answer 5: The authors would like to thank to Reviewer 3 for this suggestion. In order to give just the main conclusions of the research in Conclusion section, the authors referred additionally to the numerical results for some important mechanical properties like flexural modulus of elasticity and flexural strength, which increase by adding Calotropis fibres layers in composites investigated. As a result, the following new paragraph was added to section “4. Conclusions”:

“It must be mentioned that just the flexural modulus of elasticity and flexural strength are greater for composites DC and DTC which contain Calotropis fibres, than the ones obtained for composite D which does not contain Calotropis fibres. Moreover, it may be remarked that the chemical treatment of the Calotropis fibres led to the flexural modulus of elasticity with 12 % and with 43.5 % higher than the one corresponding to the composite DC containing untreated Calotropis fibres in longitudinal and transversal directions of the fibres, respectively. The increase of the flexural modulus of elasticity by adding treated or untreated Calotropis fibre sheets in composites is the cause of the decrease of the impact strength which varies inversely proportional to the flexural modulus of elasticity.”

Question 6: Title is not reflecting the contents of the manuscript.

The suggested title may be Investigation on Phoenix Dactylifera/Calotropis procera fibre reinforced epoxy hybrid composites

Answer 6: The title was changed, as suggested
